# A Fixed-Film Bio-Media Process Used for Biological Nitrogen Removal from Sewage Treatment Plant

**Jesmin Akter** [1,2] , **Jaiyeop Lee** [1,2] **and Ilho Kim** [1,2,*]

1   Department of Environmental Research, Korea Institute of Civil Engineering and Building Technology (KICT), Goyang-si 10223, Korea
2   Department of Civil and Environmental Engineering, University of Science and Technology, Daejeon 34113, Korea
*   Correspondence: ihkim@kict.re.kr

**Abstract:** In this study, a lab-scale fixed-film bio-media process was developed and operated to evaluate nitrogen removal from domestic sewage treatment plants. For nitrogen removal, the fixed-film bio-media process was applied in series with anaerobic, anoxic, and aerobic units in three separate reactors that were operated continuously at the same loading rates and hydraulic retention time. A biofilm separation bioreactor was developed for on-site domestic wastewater treatment and the bioreactor employed synthetic fiber modules so that the biomass could be completely attached to the media. In this paper, the performance of the fixed-film bio-media process with an average flow rate was evaluated before and after stabilization of the treatment system for nitrogen removal. The results show that the fixed-film bio-media process was successful for improved nitrogen removal from secondary and tertiary treated wastewater, with a 77% decrease in the total nitrogen discharge. Rapid nitrification could be achieved, and denitrification was performed in the anoxic filter with external carbon supplements during tertiary treated sewage wastewater. However, aeration was supplied after the stabilization process to achieve the nitrification and denitrification reaction for nitrogen removal. However, stable aeration supply could enhance nitrification at moderate temperature with benefits from complete retention of nitrifying bacteria within the system due to bio-media separation.

**Keywords:** biological nitrogen removal; bio-media process; domestic wastewater; fixed film bio-media

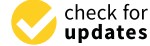



## 1. Introduction

Nutrients are essential for plants and animals; about 80% of the earth's atmosphere is made up of nitrogen. Nitrogen in wastewater discharge is undesirable for several reasons; its reduced forms expend oxygen demand in the receiving water body [1]. In wastewater, if nitrogen concentrations are high, a large amount of chlorine for its disinfection [2] is required. Nitrogen can be in wastewater in different ionized forms: ammonium ($NH_4^+$) and ammonia ($NH_3^+$); also, it depends on the concentration, pH, and temperature [3]. However, ammonia removal is accomplished through biological nitrification and denitrification processes. Ammonia turns to nitrite and then nitrate by nitrifying bacteria called nitrifiers [4]. Therefore, multiple different methods of nitrogen removal from wastewater and advanced adherent growth processes are considered attractive options for wastewater treatment. The methods include ion exchange, adsorption, reverse osmosis, and chemical processes such as active metal and catalytic methods [5–8].

Bio-media reactors replace treatment plant designs that are more efficient and perform better than traditional wastewater treatment processes [9]. Numerous studies have determined the effectiveness of attached growth systems for removing organic matter from wastewater treatment. However, the limitations of attached growth systems in bio-media processes are rarely discussed in detail due to media clogging.

Thus, to overcome the problems, one must improve the efficiency of the bio-media process. Biological nutrient removal processes are cost-effective, environmentally friendly,

and allow for the chemical treatment of wastewater. Biological systems are generally an efficient, simple, and economical way to recover or remove nutrients, remove undesirable compounds, and improve digestive quality [10].

Therefore, several methods to remove nitrogen from wastewater, along with advanced attached growth processes are considered good options for wastewater treatment. The advantages of the attached growth systems over conventional activated sludge processes include better oxygen transfer, high nitrification rate and biomass concentrations, more effective organic removal, and relatively shorter hydraulic retention time (HRT) [11,12]. In attached growth systems, most support media affect process efficiency and performance. Bio-media systems are state-of-the-art wastewater treatment processes that have the advantages of both attached and suspended growth systems. The attached growth system is based on a carrier to which biomass is attached and the properties of the media carrier are influenced by the behavior and performance of the biological treatment system for the various sizes, types, surfaces, and pores of the biological media [12]. Therefore, a medium selection is required. Selection of the appropriate bio-medium used and selection of media such as media type, media size, surface area, and media porosity are also important. The choice of biological medium will affect the ability of the biofilm and its ability to remove the biofilm. This is because there is no need to change the medium during the operation of the biotreatment process. The ideal bio-media is a material with criteria such as high specific surface area, high porosity, light weight, and very high durability [13]. Mostly, bio-media are the physical basis for biomass growth, and a few types of bio-media can be used, such as rock, glass, plastic, and wood. Here, plastic is one of the best choices for bio-media. Structured plastic media are modernizing the design and operation of biofilm reactors. Synthetic plastics are used nowadays because of their high porosity, maximum specific surface area, and light weight. These types of media are easy to carry and install; they offer clog resistance and high porosity [13]. The plastic media have a larger surface area that can allow more bacteria growth on the surface media and is more economical [14].

Therefore, the biological treatment process has been improved by combining with support media for the growth of a wide variety of microorganisms in the form of biofilm, which supports easier control and enhances efficiency [15]. This paper presents a lab-scale bio-media nitrogen removal process from domestic wastewater through a fixed-film media process.

This research aimed to remove nitrogen from domestic wastewater by a lab-scale fixed-film bio-media process. The fixed-film bio-media process was in series with anaerobic, anoxic, and aerobic units in three separate reactors operated continuously at the same loading rates and hydraulic retention time for nitrogen removal. A biofilm separation bioreactor was developed for on-site domestic wastewater treatment, and the bioreactor employed synthetic fiber modules so that the biomass could be attached to the media. Biological filtration uses the bio-media filtration technique that utilizes microorganisms to remove nutrients from the wastewater.

The two main strategies for nitrogen removal were nitrification and denitrification. Nitrification occurred in two stages. First, ammonia ($NH_3$) is converted to nitrite ($NO_2^-$) by ammonia-oxidizing bacteria (AOB), and in the second step, nitrite-oxidizing bacteria (NOB) converts nitrite to nitrate ($NO_3^-$). Denitrification is an anaerobic process. This converts $NO_3^-$ into nitrogen gas ($N_2$) by heterotrophic microorganisms that use a carbon source as an electron donor. In domestic wastewater, nitrogen comes from human waste and other organic waste. Removal of nitrogen from wastewater has become a global concern because eutrophication and nitrates are dangerous to human health. In this paper, bio-media were used for the nitrification and denitrification processes and biofilm growth is also a possibility for ammonia removal. Therefore, the specific surface area of the medium can be used for biofilm maturation and the type and texture of the medium influence nitrification.

In this study, a pilot plant, consisting of three upflow filters, was set up at a municipal sewage treatment plant and operated for 210 days with a feed of secondary-treated real sewage. The target water quality T-N in the effluent was set at less than 5 mg/L by consider-

ing the water quality of effluent in Korea. Biological nutrient removal (BNR) systems have been widely applied worldwide as excellent wastewater treatment systems due to their advantages over other systems [16]. In general, BNR systems comprise submerged media wastewater treatment reactors that combine biomass separation with oxygen biological treatment [17]. In this experiment, a fixed-film medium was used as the support for microbial biofilms. This process shows some distinct advantages over conventional activated sludge processes, including higher biomass concentrations, simplicity of operation and higher process stability, high effluent quality associated with suspended growth systems, and the diffusional barriers of the biofilm [18]. On the other hand, there is difficulty in its application when treating the sewage because of the high concentration of suspended solid (SS). During this treatment process, SS removal may cause loss of organic matter and it also decreases the carbon-to-nitrogen ratio (C/N ratio) of wastewater. Therefore, sufficient carbon needs to completely denitrify the nitrate during the nitrification process [19] and denitrification process efficiency is also quite low due to the low C/N ratio of influent wastewater [20,21]. To enhance the denitrification efficiency, post-denitrification in this process used an external carbon source to balance the C/N ratio [22,23].

In the fixed-film bio-media process, microbial growth on the surface of the medium and bio-carriers is attached throughout the reactor [9,10]. Microorganisms grow on the surface area of the bio-carrier, making the bio-media process effective; the biofilm surface area is a key design parameter [19]. A wide range of support materials, some of them organic or synthetic, for example, wood, gravel, rock, and synthetic materials: ceramic, nylon, and polyethylene [23,24], have advantages, as lower density can influence air flows and hydraulic speeds and can have an impact on the mass and oxygen transfer [25]. As well as resistance, there is less volume requirement, no recycling or backwashing, and no mechanical intervention in the case of load fluctuations.

In the system on a biofilm basis, biofilms form diverse groups of microorganisms such as bacteria, fungi, yeasts, *etc.*, and microorganisms are protozoa, worms, insect larvae, *etc.*, and extracellular polymers (EPS). The character of biofilms is usually grayish and muddy [26]. A slime layer of microorganisms is formed on the surface of the medium. Microorganisms multiply and the thickness of the slime layer also increases [27]. Microorganisms form a biofilm on the media, enhancing the performance of the activated sludge system and increasing the treatment capacity [28]. Adsorbed organic matter that is decomposed before reaching the microbes near the surface of the medium also enters the endogenous growth stage due to starvation. Microorganisms lose their ability to stick to the media surface due to organic matter that has not reached the inner layer of the bio-media. When the wastewater washed the slime from the media, a new layer of slime started to grow. The situation in which the mucus layer is lost is called sloughing and is primarily a function of the organic and hydraulic loading of the filter. However, growing biofilm, within the inside of the bio-media, degrades dissolved pollutants in the wastewater [27]. To treat the wastewater, it is generally applied intermittently or continuously over the media, via upflow or downflow. Typically, a bio-media process has two or three phases, depending on the concentrations of nutrients. During the treatment process, organic matter and other water components diffuse into the biofilm by biodegradation. These processes are usually aerobic, which means microorganisms require enough oxygen for their metabolism, and oxygen can be supplied to the biofilm by blowers. The process performance depends on the microorganisms' activity; also, the crucial influencing factors are the water composition, the biofilter hydraulic loading, the type of media, the feeding strategy (percolation or submerged media), and the age of the biofilm, temperature, aeration, *etc.*

## 2. Materials and Methods

### 2.1. Laboratory-Scale Experimental Setup

The experimental device, shown in Figure 1, was installed at a sewage treatment plant in South Korea. In this experiment, the effluent from the tertiary and secondary sedimentation tank was used as the target water of treatment. The experimental devices

consisted of three upflow biological filters. The reactor volume was 2.5 L, and the filtration media used were fixed types made of plastic fiber. The length of the media in each filter was 20 cm, the volume was 7.5 cm$^3$, and the density was 0.12 g/cm$^3$. Three layers of media were tied in the middle of each reactor. The system was operated in two-step and three-phase cycles based on the sedimentation tank's influent water. The first phase's duration was eight weeks, and the raw water was from the tertiary sedimentation tank. In phase II, when the secondary effluent was applied, additional methanol was added as an external carbon source to the storage tank. Phase II was operated under anoxic conditions for denitrification. For denitrification, to secure enough organic material is essential as a hydrogen donor. So, methanol was added to the storage tank as an external carbon source. In phase III, the first reactor aerated the removal of residual methanol and the recovery of dissolved oxygen (DO). A sketch of the lab-scale bio-media fixed-film reactors is shown in Figure 1.

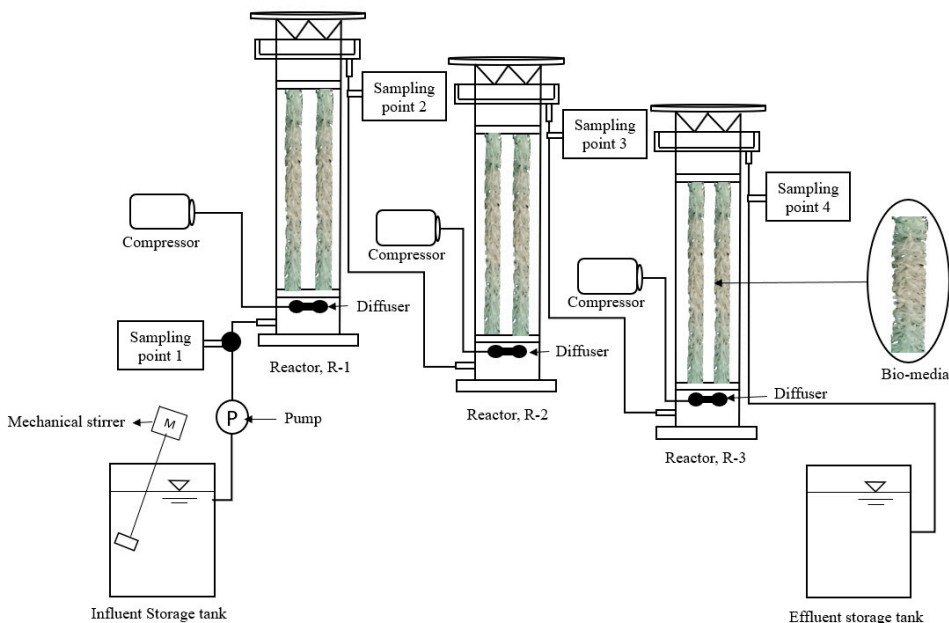

**Figure 1.** Lab-scale bio-media fixed-film reactors.

### 2.2. Operational Condition

Continuous treatment was started on 4 March 2020 and continued until 16 September 2020. At the beginning of the operation, activated sludge was added to each reactor as a microorganism source. Operating conditions are listed in Table 1. Three different periods have been investigated throughout this research. The water quality parameters are shown in Table 2. In this study, both nitrification and denitrification steps were incorporated with the two-step cycle shown in Figure 2c. Phases I and II were start-up periods for stabilization in step 1. Due to high DO concentration of around 5–7 mg/L in phase I and II, all the reactors operated without aeration. In phase III, nitrification and denitrification occurred in reactors R1, R2, and R3 under oxic and anoxic conditions. Each phase of the experiment was carried out in sequence by switching the flow directions, as shown in Figure 2a,b. In step 1, the anoxic reactor R1 was filled with the influent water. The first nitrification without interfering with the organic matter occurred in the 2nd reactor R2, and the third reactor R3 functioned as denitrification. When changing the flow direction in the second step, the biomass previously adsorbed on the media was used as the substrate for the nitrification reaction. In this study, denitrification in the 2nd and 3rd anoxic reactors occurred by switching the flow direction and this is an essential distinction of this BNR system. In addition, independent nitrification can be led by the separation of heterotrophs and autotrophs and thus the configuration of the BNR system would enhance its nitrogen removal.

**Table 1.** Operational conditions.

| Operating Conditions | Unit | Weeks of Operating | | | | |
|---|---|---|---|---|---|---|
| | | Phase I | | Phase II | | Phase III |
| | | Week 1–3 | Week 4–8 | Week 8–17 | Week 18–21 | Week 22–28 |
| Flow rate | mL/min | 20 | 10 | 10 | 10 | 15 |
| HRT | h | 2 | 4 | 4 | 4 | 3 |
| Methanol addition | mg/L | - | - | 10 | 10 | 10 |
| Inflow water | | Tertiary | Tertiary | Secondary | Secondary | Secondary |
| DO | mg/L | 5–7 | 3–7 | 2–3 | 2–3 | 2–3 |

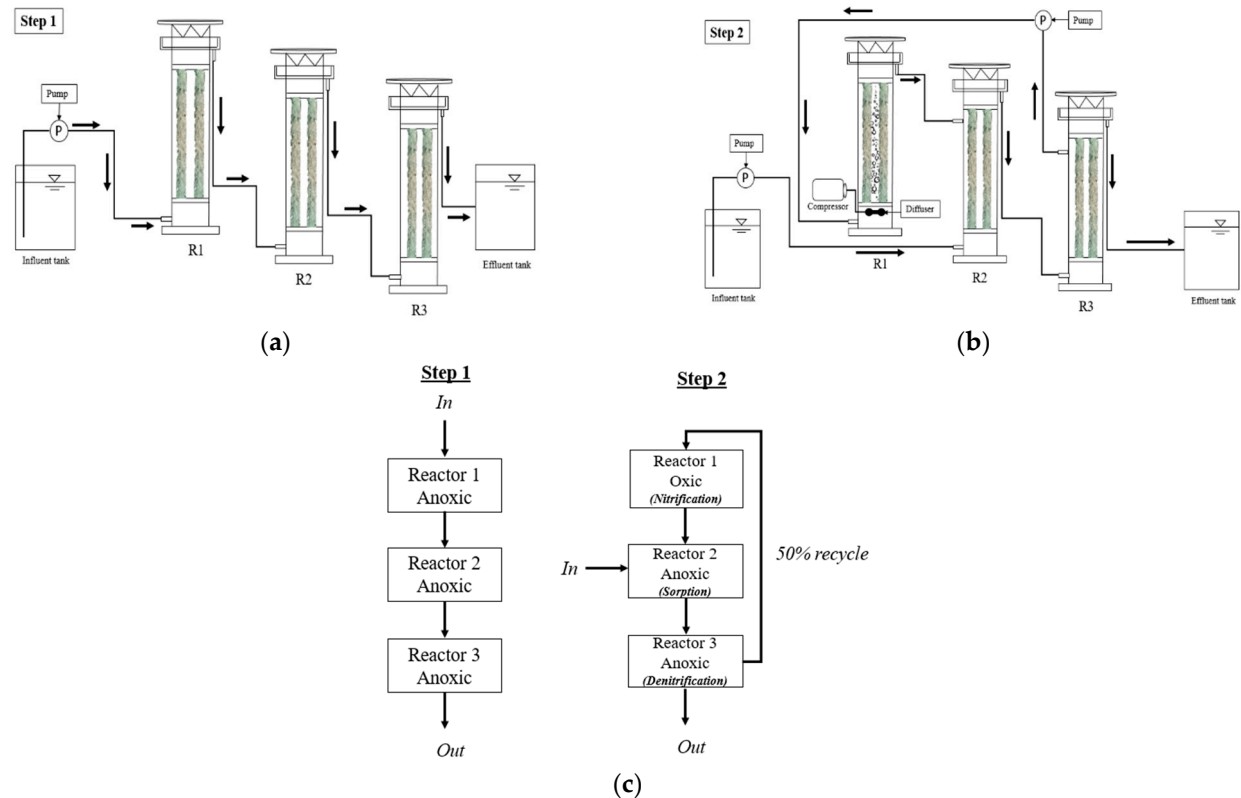

**Figure 2.** (**a**,**b**) Schematic of BNR system in step 1 (upper) and step 2 (bottom) depending on the direction of influent flow; (**c**) operational step of the three-reactor BNR process.

**Table 2.** Properties of inflow wastewater.

| | Parameters | DO (mg/L) | pH | T (°C) | T-N (mg/L) | $NH_3^+$-N (mg/L) | $NO_3^-$-N (mg/L) | $NO_2^-$-N (mg/L) | Org-N (mg/L) | COD (mg/L) |
|---|---|---|---|---|---|---|---|---|---|---|
| Phase I | Inflow (Tertiary Water) | 2.5–6.5 | 6.3–7.3 | 9.50–18.00 | 6.50–13.60 | 0.01–1.10 | 0.40–5.60 | 1.00–3.00 | 2.20–7.40 | 8.30–14.90 |
| Phase II | Inflow (Secondary Water) | 2.0–3.0 | 6.5–7.5 | 15.00–32.00 | 4.40–13.80 | 0.01–1.00 | 1.00–4.00 | 0.01–3.00 | 0.10–11.69 | 8.00–29.00 |
| Phase III | Inflow (Secondary Water) | 2.0–3.0 | 6.0–7.0 | 22.00–32.00 | 2.30–6.70 | 0.01–1.00 | 0.30–2.40 | 0.01–0.35 | 1.81–5.00 | 8.00–39.00 |

*2.3. Sampling and Analysis*

Samples were collected using 100 mL plastic bottles from all the sampling points of each reactor three times every week for up to seven months. Analysis of soluble ammonia ($NH_3$-N), nitrate ($NO_3$-N), nitrite ($NO_2$-N), soluble COD, total nitrogen (T-N), and suspended solids (SS) was conducted by standard methods of C-MACH and HACH kits using spectrophotometer DR-5000. The samples were analyzed immediately after sampling and using filtration through 0.45 μm filter papers. Temperature, dissolved oxygen (DO),

and pH were checked in each reactor every sampling time immediately before collecting samples. The measurements of all DO and pH were carried out with a DO (Model S-6120) and pH meter (Model HM-501), respectively.

## 3. Results

In this study, a fixed-film bioreactor with attached activated sludge was used as a reference system. Throughout this experiment, an anoxic system (HRT 2 h) and an anaerobic system (HRT 4 h) based on activated sludge with a fixed film and their results were compared. A total of 60 days of operation were required to reach the steady-state conditions for bioreactors; after this period, the main sampling and analyses were performed. The results are presented according to the phase in the following sections.

### 3.1. Phase I

In the first stage of this study, throughout the experiment of phase I, DO was above 6 mg/L. The average concentration of DO in this study was 2–3 mg/L. These levels of DO were adequate for efficient nitrification performance. In phase I, the water temperature was between 9 and 18 °C, and the pH value was between 7.0 and 7.5. The ammonium concentrations and removal efficiencies were determined over time through the first stage. The mean concentration of ammonium was as follows: in reactor R1 (0.35 mg/L), in R2 and R3 (0.25 mg/L and 0.32 mg/L, respectively). The removal percentages of $NH_3$-N reached up to 35%.

The nitrate in the effluent was produced by nitrifying bacteria that oxidize ammonium. The mean concentration of $NO_3$-N in the first stage of the influent was only around 3.31 mg/L, but it was similar in the effluent of all the reactors. On the other hand, the nitrite concentration during phase I was around 1–3 mg/L in all the reactors. However, in this phase, the removal efficiency of ammonia, nitrate, and nitrite was very low. Initially, the T-N concentrations in the first stage influent were 9.63 mg/L and effluent was around 8.39 mg/L in all three reactors, which indicates very low nitrogen removal at this early stage of the experiments. The COD concentration of the first stage influent was around 12.5 mg/L. This value was reduced during the 1st stage to 11.7 mg/L, 11.6 mg/L, and 11.8 mg/L for the effluent of R1, R2, and R3, respectively. The concentration of Org-N in influent was on average 4.1 mg/L. This showed that the concentration of Org-N in the effluent of all reactors decreased by nearly 2.5 to 3.00 mg/L.

### 3.2. Phase II

The average concentration of DO in this phase was 2–3 mg/L. These levels of DO were adequate for efficient nitrification performance. In phase II, the water temperature was between 15 and 33 °C, and the pH value was between 6.4 and 7.4. The ammonia nitrogen concentrations in the second stage influent were 0.16 mg/L. The mean concentration of ammonium was as follows: in reactor R1 (0.12 mg/L), R2 and R3 (0.13 mg/L and 0.11 mg/L), respectively. The removal percentages of $NH_3$-N reached up to 40%. As shown in Figure 3, for the mean concentration of $NO_3$-N in this stage, the influent was only around 2.26 mg/L. The mean concentration of nitrate was as follows: in reactor R1 (1.61 mg/L). Therefore, in this phase, the removal efficiency of ammonia nitrogen, nitrate, and nitrite did not significantly increase in the effluent. In this phase, the T-N concentrations in the influent had 7.24 mg/L, and the effluent of all the reactors had 5.59 mg/L, 5.75 mg/L, and 5.09 mg/L, respectively. The removal efficiency of total nitrogen was 30%, which indicates a relatively low nitrogen removal performance at this stage of the experiments.

The mean COD concentration in the second stage effluent was around 17.6 mg/L. This higher value specified the addition of a carbon source to the second phase. The influent showed an increase of COD value and promoted the removal of COD as well during this stage. The addition of an external carbon source enhanced the activity and growth of denitrifying bacteria. During this stage of the experiments, the COD decreased

to 15.9 mg/L for the effluent. The concentration of Org-N in the influent was an average of 4.50 mg/L and in the effluent of all reactors, it decreased by nearly 3.00 mg/L.

### 3.3. Phase III

The average concentration of DO in this phase was 1–3 mg/L. In III, the water temperature was between 22 and 32 °C, and the pH value was between 6.0 and 7.0. The ammonia nitrogen concentration in phase III influent was 0.12 mg/L. The efficiencies of ammonium in the effluent of phase III were determined. The concentration of ammonium decreases up to 0.01 mg/L. The removal percentages of $NH_3$-N reached 91.67%. For the mean concentration of $NO_3$-N in this stage, the influent was only around 1.0 mg/L. The mean concentration of nitrate was as follows: in reactor R1 (1.14 mg/L), R2 and R3 (1.02 mg/L and 1.06 mg/L, respectively). However, the nitrate concentration during phase III was low and around 0.20–0.30 mg/L in all the reactors. The removal percentages of $NO_3$-N and $NO_2$-N reached 76.66 and 95.62%. Therefore, in this phase, the removal efficiency of ammonia nitrogen, nitrate, and nitrite did increase in the effluent.

In phase III, the average T-N concentrations in the influent were 4.6 mg/L, and the effluent of all the reactors was around 3.8 mg/L, respectively. The removal efficiency of total nitrogen reached up to 70%, which indicates a relatively moderate nitrogen removal performance at this stage of the experiments.

The mean COD concentration in phase II and phase III effluent was similar, around 17.6 mg/L. In this stage of the experiments, the COD decreased up to 16.3 mg/L for the effluent. The concentration of Org-N in the influent was an average of 3.30 mg/L. It was observed that the concentration of Org-N in the effluent of all reactors decreased by nearly 2.50 mg/L.

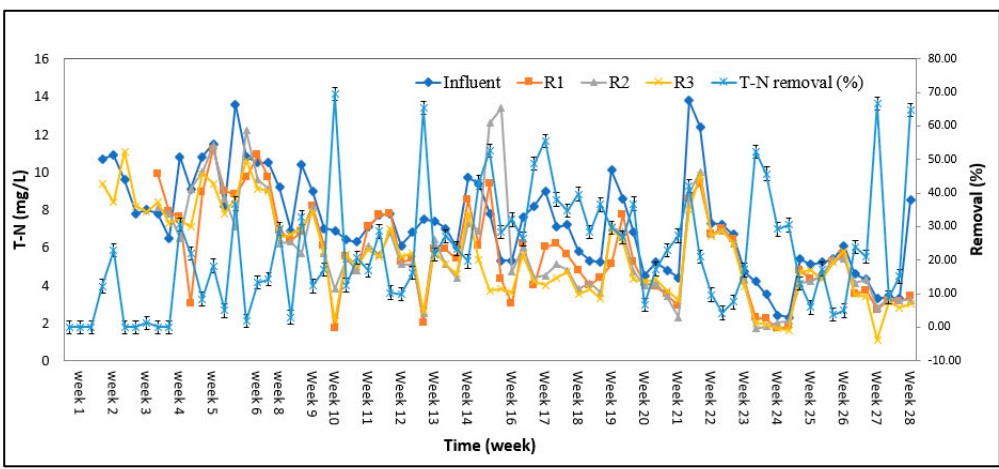

**Figure 3.** Total nitrogen concentration and removal efficiency (%).

### 3.4. Full-Scale

The removal effectiveness of total nitrogen reached up to 70% and achieved permissible effluent nitrogen concentration lower than 5 mg/L. The COD/N ratio values in treated wastewater changed in the range from 1.20 to 21.50 mg. The research showed that when the COD/N ratio was lower, the concentration of $NO_3$-N was higher, and the nitrification process was stable. Higher parameter values destabilized the process in the last phase (Figure 4a). The total nitrogen concentration of effluent ranges between 2.30 and 13.80 mg/L. Changes in concentrations of total nitrogen and nitrogen removal efficiency at all phases are shown in Figure 4b,c. For the biological process, understanding of the microbial structure that conforms to the biological system is needed for its proper function [1]. The functions are accumulated with three factors; the load of organic matter, the ammonium concentration, and the oxygen concentration. Organic load controls nitrification rate and it should be as low as possible. For the proper nitrification process, the DO concentration

in the aerobic reactor should be sufficiently high to penetrate through the outer layer of oxygen-consuming heterotrophs and into the nitrifying bacteria [29].

Figure 4d shows the correlation between DO concentrations and ammonium loading rates in the oxic reactor. As the figure shows, the concentration of DO in the aerobic reactor decreased with increasing ammonium loading rates.

The data have been calculated based on lab-scale influent and effluent $NO_x$-N concentrations, the reactor (R3), and flow rate per day. As indicated in Figure 4c, the denitrification rate is enhanced with increasing $NO_x$-N loading. The $NO_3$-N and $NO_2$-N loading rate per unit of media length has been calculated and is shown in Figure 4e,f.

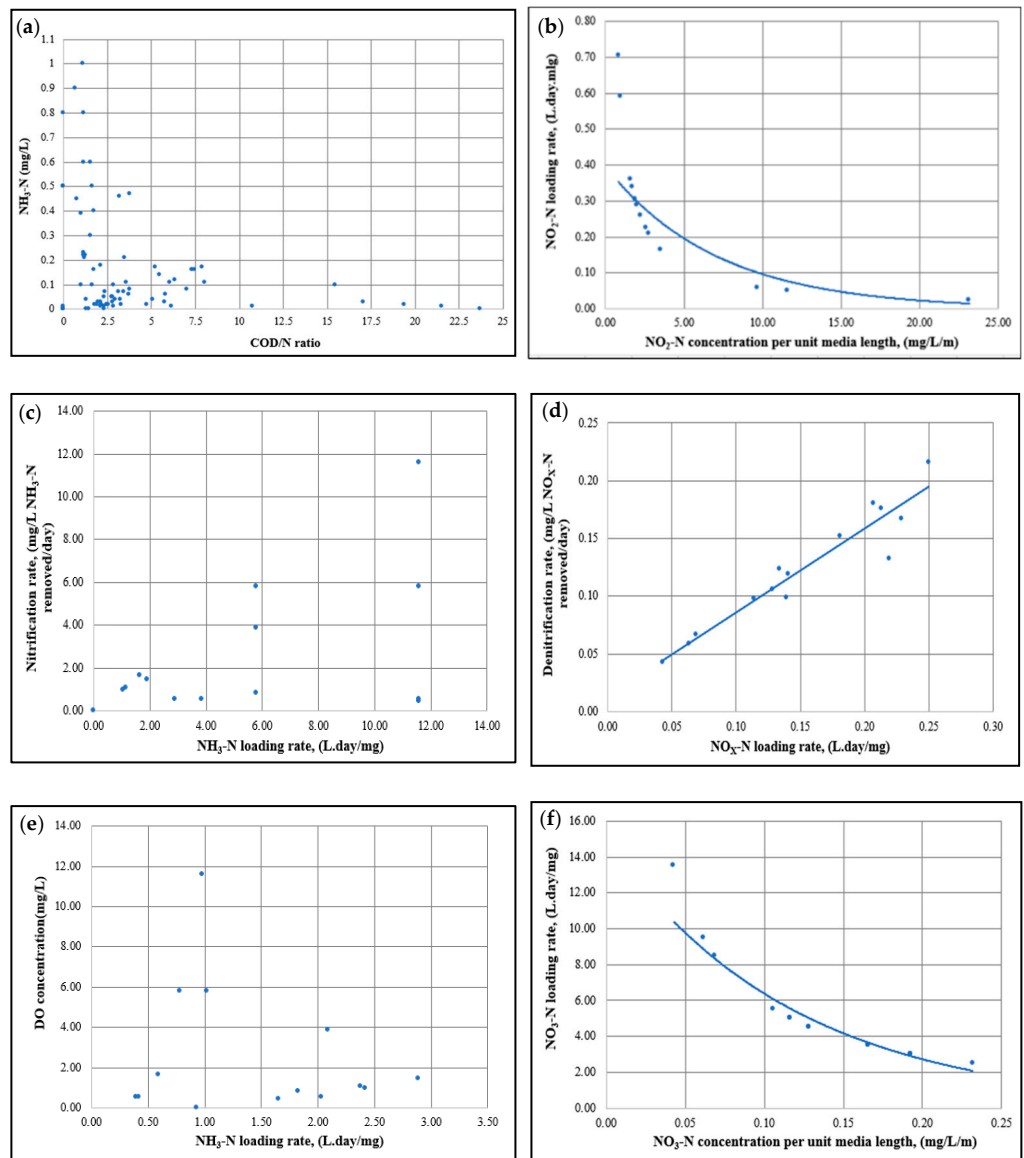

**Figure 4.** (**a**) COD/N ratio in treated wastewater effect on $NH_3$-N concentration. (**b**) Nitrification rate versus ammonia loading rate in the oxic reactor. (**c**) Denitrification rate versus $NO_x$-N loading rate. (**d**) Dissolved oxygen (DO) ammonia loading rate. (**e**) $NO_3$-N loading rate versus $NO_3$-N concentration per unit media length. (**f**) $NO_2$-N loading rate versus $NO_2$-N concentration per unit media length.

## 4. Discussion

There are different biological technologies that have been developed for nitrogen removal; for example, a single reactor system for high ammonium removal over nitrite, anaer-

obic ammonium oxidation (ANAMMOX), and simultaneous nitrification-denitrification (SND). The nitrogen removal performances are enhanced due to biofiltration through a submerged medium that serves two purposes. First, the following occur: biological conversion of organic contents by the biomass attached to the fixed support medium surface and physical retention of suspended particles. Therefore, it is also very important to perform anaerobic treatment of sewage water with high carbon content to subsequently promote the aerobic system nitrification-denitrification and reduce the necessity of DO with heterotrophic bacteria [30]. A high C/N ratio influent can also negatively influence the abundance of nitrifying bacteria and nitrification process efficacy, due to the domination of heterotrophs. From the literature review, total nitrogen (TN) removal rate reached 77% at a C/N ratio of 19.5 and a rate of 87% at a ratio of 7.7 using the simultaneous nitrification-denitrification (SND) process [26]. Therefore, this is one promising technology for the removal of ammonium and other nitrogenous compounds in concentrations higher than 250 mg N/L [31,32]. In this study, the results showed removal of nitrogen at the HRTs of 2 h and 4 h, respectively. The nitrogen in effluent at the HRT of 4 h was in the range of 2.3–13.80 mg/L (average of 6.01 mg/L). Figure 3 shows the T-N concentration of effluents, and the average effluent concentration of ammonia nitrogen was 0.15 mg/L. The removal efficiencies of ammonia nitrogen were 76.9% and 85.7% for the HRTs of 4 h and 2 h, respectively. Comparatively relevant removal of nitrogen was observed and the process application of attached growth with fixed-film bio-media shows relative performance. Moreover, the configuration of the system allowed for independent nitrification and this would be the reason for an increase in the nitrification rate within short HRT. The $NO_x$-N concentration of the final effluent reached at the HRTs of 4 h and 2 h was 22.2 mg/L and 25.28 mg/L on average, respectively. However, the effluent $NO_x$-N was low and the TCOD/TKN ratio of the influent at the HRTs of 4 h and 2 h was 7.74 and 1.12, respectively. In the operation of this three-stage BNR system, the effluent with low $NO_x$-N concentration could be benefited by its application.

## 5. Conclusions

In conclusion, we conducted the evaluation of this proposed three-stage BNR system for enhancing nitrogen removal in municipal wastewater treatment. Based on the results of the experimental tests, the lab-scale BNR system with fixed-film plastic bio-media proposed in this study removed organics and nutrients efficiently. However, there were some limitations over long HRTs and due to the addition of carbon sources. This study shows that the removal efficiency of TN from treated wastewater by a three-stage BNR process could reach up to 77%. However, the proposed system was less effective in denitrification due to low C/N ratio, but the fixed-film bio-media minimize the organic loss and lead to better use of bio-attached organics to enhance the denitrification reaction. Wastewater temperature and the DO concentration also affect the process efficiency. The most favorable conditions were when the wastewater temperature was above 15 °C and the DO concentration was 2–3 mg/L. Overall, the removal of nitrogen was enhanced by changing the flow direction of the influent. The reactors employed three functioning reactors for the role of nitrification and denitrification.

**Author Contributions:** J.A.: sample collection, sample analysis, writing, and original draft preparation. J.L.: conceptualization, investigation, methodology, and resources. I.K.: conceptualization, investigation, methodology, resources, supervision, and funding acquisition. All authors have read and agreed to the published version of the manuscript.

**Funding:** This research was supported by the Korea Institute of Civil Engineering and Building Technology (KICT), research project entitled "Development of practical technology for resource and energy recycling system using organic waste biogas", project number #20220081-001.

**Conflicts of Interest:** The authors declare no conflict of interest.

## Abbreviations

| | |
|---|---|
| AOB | Ammonia-oxidizing bacteria |
| NOB | Nitrite-oxidizing bacteria |
| COD | Chemical oxygen demand |
| TCOD | Total chemical oxygen demand |
| $NO_2$-N | Nitrite nitrogen |
| $NO_3$-N | Nitrate nitrogen |
| T-N | Total nitrogen |
| TKN | Total Kjeldahl nitrogen |
| $NH_3$-N | Ammonia nitrogen |
| BNR | Biological nutrient removal |
| SS | Suspended solid |
| DO | Dissolved oxygen |

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
