# Peer review of "A Fixed-Film Bio-Media Process Used for Biological Nitrogen Removal from Sewage Treatment Plant"

_nitrogen, doi:10.3390/nitrogen3030034_

Round 1
Reviewer 1 Report
This paper describes a lab-scale nitrogen removal process from domestic wastewater through a fixed-film media process, in three separate reactors operated continuously for nitrogen removal.
Numerous grammatical and editing errors throughout the text. I only show the first seen
Line 33: rates. [1].
Line 53: grown. [5].
Line 61: , high porosity or high porosity,
The introduction is very difficult to read and understand, dealing with the different topics in a disorderly way.
The 17 references that appear are all in the introduction.
Legend of figure 1 not explanatory enough
Fig 3. It does not explain the number of times the experiment is repeated, nor the mean deviation, nor are there error bars.
Several times is cites the Fig. 5, which does not exist in the paper
In the discussion the only thing that is done is to summarize agian and put back the results obtained. A sign that the discussion is totally insufficient is that there is not a single citation to which reference is made.
Author Response
Thank you very much for your comments that helped us improve this manuscript.
This paper describes a lab-scale nitrogen removal process from domestic wastewater through a fixed-film media process, in three separate reactors operated continuously for nitrogen removal.
- Numerous grammatical and editing errors throughout the text. I only show the first seen
Line 33: rates. [1].
Line 53: grown. [5].
Line 61: high porosity or high porosity,
Response: Checked the grammatical and editing errors and corrected them accordingly.
- The introduction is very difficult to read and understand, dealing with the different topics in a disorderly way.
The 17 references that appear are all in the introduction.
Response: The introduction has been revised, and more references added.
Legend of figure 1 not explanatory enough
Response: Revised accordingly.
- Fig 3. It does not explain the number of times the experiment is repeated, nor the mean deviation, nor are there error bars.
Response: Fig 3. has been revised and error bars are provided.
- Several times is cites the Fig. 5, which does not exist in the paper
Response: Revised accordingly.
- In the discussion the only thing that is done is to summarize again and put back the results obtained. A sign that the discussion is totally insufficient is that there is not a single citation to which reference is made.
Response: Revised and checked accordingly.
Reviewer 2 Report
In the article submitted for review a lab-scale fixed film bio-media process was developed and operated to evaluate nitrogen removal from domestic sewage treatment plant. For nitrogen removal, the fixed film bio-media process was applied in series with anaerobic, anoxic, and aerobic units in three separate reactors that were operated continuously at same loading rates and hydraulic retention time.A bio-film separation bioreactor developed for on-site domestic wastewater treatment and the bioreactor employed synthetic fiber modules so that the biomass could be completely attached with the media. al. The results show that the fixed-film bio-media process was successful for improved nitrogen removal from secondary and tertiary treated wastewater, with a 70% decrease on the total nitrogen discharge. This is very good information for technologists developing technologies for treating wastewater from households.
However, the article needs to be improved. This version is chaotic.
· Please use the citation record correctly. A period appears before the quotation. Please delete them, e.g. on lines: 33, 53, 115, etc.
· When double quoting, a space appears once, eg in line 100, and in other cases it is not there, for example in line 108. Please make it uniform throughout the text.
· Please standardize the units, cm3 appears in the text, and ml appears in the table incorrectly (mL).
· In chapter 2.4, please write the second part in lower case.
· Figure 3 needs improvement. Please put only part a or only part b. If there are both parts, please change the caption under the drawing because part b does not include the%.
· In chapter 3.4, figure 5 is mentioned several times. There is no such figure. Please correct this.
· Please correct the description of figure 4. Please decide whether you use periods or commas between the signs a, b, c etc. I suggest commas.
· The notation of the cited literature requires standardization in accordance with the requirements contained in the guide for authors.
· The cited literature comprises only 17 items. It is a little bit too little.
Thank you for considering my opinion. I encourage the authors to continue working on improving the manuscript.
Author Response
Thank you very much for the review. We have made revisions accordingly.
- Please use the citation record correctly. A period appears before the quotation. Please delete them, e.g. on lines: 33, 53, 115, etc.
Response: Revised accordingly.
- When double quoting, a space appears once, eg in line 100, and in other cases it is not there, for example in line 108. Please make it uniform throughout the text.
Response: Corrected accordingly.
- Please standardize the units, cm3 appears in the text, and ml appears in the table incorrectly (mL).
Response: Corrected accordingly.
- In chapter 2.4, please write the second part in lower case.
Response: Corrected accordingly.
- Figure 3 needs improvement. Please put only part a or only part b. If there are both parts, please change the caption under the drawing because part b does not include the%.
Response: Fig 3. has been revised and corrected accordingly.
- In chapter 3.4, figure 5 is mentioned several times. There is no such figure. Please correct this.
Response: Corrected accordingly.
- Please correct the description of figure 4. Please decide whether you use periods or commas between the signs a, b, c etc. I suggest commas.
Response: Corrected accordingly.
- The notation of the cited literature requires standardization in accordance with the requirements contained in the guide for authors.
Response: Revised accordingly.
- The cited literature comprises only 17 items. It is a little bit too little.
Response: More references have been added.
Thank you for considering my opinion. I encourage the authors to continue working on improving the manuscript.
Reviewer 3 Report
This problem is relevant to the journal scope. The content is well presented, concept and aim are clearly defined.
I suggest the minor revision of the manuscript.
Remarks
1. Please cite more papers from this journal in the last two years on a similar topic to this research.
2. Please compare your results with other literature data.
3. The main remark: Please emphasize the novelty side(s) of your manuscript.
4. Please describe the conditions of the analytical methods.
5. Please describe the circumstances of the sampling in more detail.
6. In Figure 4, please avoid the straight fitting in the case of c and e parts.
Author Response
Thank you very much for your valuable comments that helped us improve this manuscript.
Remarks
- Please cite more papers from this journal in the last two years on a similar topic to this research.
Response: More paper has been cited accordingly.
- Please compare your results with other literature data.
Response: Revised and compared accordingly.
- The main remark: Please emphasize the novelty side(s) of your manuscript.
Response: Revised accordingly.
- Please describe the conditions of the analytical methods.
Response: Analytical methods condition has been described.
- Please describe the circumstances of the sampling in more detail.
Response: The sampling in detail is described.
- In Figure 4, please avoid the straight fitting in the case of c and e parts.
Response: Revised accordingly.
Round 2
Reviewer 1 Report
I value the effort made by the authors who have correctly followed my suggestions and I consider that the paper in its current state can be accepted for publication in Nitrogen.